# Understanding Data Temporality Impact on Large Language Models Pre-training

Romain Fabre [* 1]   Hippolyte Pilchen [* 1]   Franck Signe Talla [1]   Patrick Perez [1]   Edouard Grave [1]

## Abstract

Large language models (LLMs) are typically trained on shuffled corpora, yielding models whose knowledge is frozen at train time and whose temporal grounding remains poorly understood. In this work, we study the impact of pre-training dynamics on the acquisition of time-sensitive factual knowledge, focusing specifically on data ordering. Our main contributions are twofold. First, we introduce a comprehensive benchmark of over 7,000 temporally grounded questions and an evaluation protocol that enables analysis of whether models correctly associate facts with their corresponding time periods. Second, we pretrain 6B-parameter models on temporally ordered Common Crawl snapshots and compare them against standard shuffled pre-training. Our results show that sequentially trained models match shuffled baselines on general language understanding and common knowledge while consistently exhibiting more up-to-date and temporally precise knowledge. Temporally ordered pre-training yields improved factual freshness, while shuffled pre-training peaks on older data, possibly due to increased factual repetition. These findings, along with the release of our code,[1] checkpoints and datasets,[2] provide a foundation for future research on continual learning for LLMs.

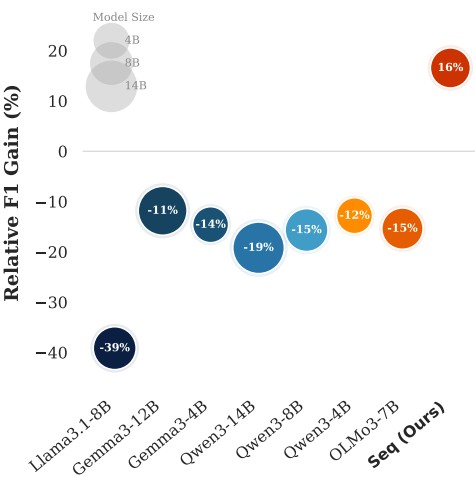

*Figure 1.* **Yearly temporal knowledge with Kairos.** Relative gains in F1 score on KairosQA between the 2020–2021 and 2023–2024 periods for our sequentially pre-trained model versus other open-source base models (ordered by their release date with the most recent at the right). These results highlight that even for recently released open-source base models, shuffled pre-training leads to a temporal delay in knowledge; performance decays when querying recent facts, even those preceding the training cut-off. Conversely, sequential pre-training represents a significant step toward developing more up-to-date models.

2020), usually made of webpages, scientific PDF documents and code repositories. This stage lays the foundation for the knowledge ultimately stored in the model's parameters.

In practice, pre-training corpora are constructed from filtered versions of multiple Common Crawl (CC) snapshots (Penedo et al., 2024). Although the filtering recipes used to build these datasets are relatively consistent across works, combining quality filtering (Touvron et al., 2023; Li et al., 2024) with deduplication (Lee et al., 2022), the ordering of data during pre-training remains under-explored. Yet, data ordering may play an important role in shaping the factual knowledge encoded in model parameters. A well-known limitation of current training pipelines is that a model's knowledge becomes effectively frozen once training is complete: LLMs cannot reliably answer questions about events occurring after the temporal cutoff of their training data. Moreover, models are often more accurate on questions tar-

## 1. Introduction

The training of large language models (LLMs) is commonly organized as a multi-stage pipeline comprising pre-training, mid-training, and post-training. During pre-training, the model is trained on vast amounts of data (Brown et al.,

---

[*]Equal contribution [1]Kyutai, Paris. Correspondence to: Hippolyte Pilchen <hippolyte.pilchen@kyutai.org>.

*Proceedings of the 43rd International Conference on Machine Learning*, Seoul, South Korea. PMLR 306, 2026. Copyright 2026 by the author(s).

[1]https://github.com/kyutai-labs/kairos
[2]Sequential Helium 6B & KairosQA

geting dates several years before the cutoff than on those near the cutoff itself (Zhao et al., 2024), revealing a gap between the training dataset horizon and the model's effective knowledge horizon. While recent work has explored continual learning for incorporating new facts (Li et al., 2025) and methods for improving temporal alignment (Park et al., 2025; Zhao et al., 2024), the impact of pre-training dynamics—particularly data sampling order—on the temporal distribution of knowledge remains poorly understood.

To investigate this question, we design an experimental framework that compares pre-training on temporally ordered data with pre-training on randomly shuffled data. Specifically, we pretrain 6B-parameter language models on either filtered sequential CC snapshots or shuffled versions of the same data, and compare their performance at a fixed token budget across a range of evaluations. We first assess language modeling and common knowledge of the resulting models using the OLMES benchmark (Gu et al., 2024). We then evaluate the models' temporal factual knowledge by constructing KairosQA,[3] a question–answering (QA) dataset of temporally sensitive facts extracted from Wikidata.[4] This dataset is designed to assess whether models encode factual information associated with specific years. Particular care is taken to formulate meaningful questions, ensuring that the evaluation measures temporal alignment rather than merely capturing cases where the model lacks the relevant knowledge. To broaden our analysis, we tested another existing temporal evaluation benchmark (Zhao et al., 2024). From these experiments, we derive empirical insights into how factual knowledge is acquired during pre-training and demonstrate the advantages of temporally ordered pre-training over shuffled training. While both approaches yield comparable performance on general-purpose language tasks, sequential pre-training consistently leads to more temporally up-to-date knowledge as underlined in Fig. 1.

Here is a summary of our main contributions:

- We design a controlled yet realistic pre-training setup in which models are trained on temporally sequential data, and we plan to release intermediate yearly checkpoints to enable further research on improving factuality and reducing forgetting in LLMs.

- We introduce a benchmark based on a time-sensitive question–answering dataset, enabling the evaluation of temporal knowledge across a diverse set of tasks.

- We analyze the behavior of intermediate training checkpoints and compare them against carefully chosen shuffled data checkpoints as baselines to study the effects of sequential pre-training.

---

[3] *Kairos* is the ancient Greek word for "time".
[4] https://www.wikidata.org/

## 2. Pre-training Sequential Models

### 2.1. Data

We construct our training corpus from multiple Common Crawl snapshots processed through a rigorous multi-stage filtering pipeline using an open-source repository, dactory.[5] We first extract plain text from HTML, discarding documents falling outside a character count range of $[500, 10^6]$. Using fastText,[6] we perform language identification, retaining documents in 24 European languages with confidence scores exceeding $0.85$.

To mitigate redundancy, we apply a Bloom filter to detect duplicate lines and discard paragraphs where novel content constitutes less than $20\%$. For quality control, we train language-specific fastText classifiers across seven domains to compute a weighted aggregate quality score. We assign domain weights as follows: Books ($1.0$), Wikipedia and Lifestyle ($0.8$), STEM and Popular Content ($0.6$), and Scientific/Humanities ($0.2$). Documents are retained if their weighted score exceeds a threshold of $0.2$, with stochastic sampling over domain contributions to avoid overly aggressive filtering. Finally, we filter degenerate text by removing documents with an $n$-gram repetition rate exceeding $0.25$ or an anomalous long-word proportion exceeding $0.1$.

### 2.2. Baseline Model

To isolate the effect of temporal ordering, we train a baseline model on a *globally shuffled* version of the dataset. This ensures that any performance divergence is attributable strictly to data curriculum rather than model capacity.

The baseline is a 6B-parameter Transformer decoder with 32 layers, 32 attention heads, and a hidden dimension of 4096. The architecture incorporates Grouped-Query Attention with 4 key-value heads, RoPE positional embeddings, and SwiGLU activations. Training proceeds for $6 \times 10^5$ steps with a global batch size of 4.2M tokens and a context length of 4096, totaling 2.5T tokens. We utilize the AdamW (Loshchilov & Hutter, 2019) optimizer with a Warmup-Stable-Decay scheduler, peaking at a learning rate of $10^{-3}$.

To ensure rigorous convergence comparisons at intermediate stages, we employ a branching cooldown strategy. Rather than evaluating the main branch directly, checkpoints are finalized by branching off the main run and applying a $30,000$-step cosine decay to a learning rate of $10^{-4}$.

The baseline corpus comprises Common Crawl snapshots spanning 2020 to 2024. While these snapshots implicitly contain persistent historical content (e.g., pages created in 2018 that remain online), the global shuffling ensures the

---

[5] https://github.com/kyutai-labs/dactory
[6] https://fasttext.cc

model views all temporal contexts simultaneously. This configuration represents a standard static pre-training regime, treating the corpus as a timeless pool of information.

## 2.3. Sequential Model

In the sequential experiment, we retain the baseline architecture and hyperparameters but process snapshots in strict chronological order. The sequential curriculum extends from early 2018 through 2025. While the baseline encounters historical content implicitly, the sequential model explicitly utilizes the 2018–2019 period to establish initial linguistic capabilities and world knowledge in their original temporal context. This allows the model to stabilize its representation of the "pre-2020" world before adapting to the distribution shifts present in later snapshots.

We acknowledge a temporal asymmetry in our experimental design: our shuffled baseline was pre-trained prior to the initiation of this project and spans a 2020–2024 window, which reflects standard practices in open-source LLM development where models are conventionally trained on shuffled, time-aggregated text corpora (Touvron et al., 2023; Olmo et al., 2026). For the sequential pipeline, we deliberately expanded the training range to include a wider multi-year timeline. As demonstrated by our snapshot quality analysis in Section 5.5, the older 2018–2019 data yields inherently lower baseline model quality and does not grant an artificial performance advantage to the sequential curriculum over the shuffled base.

The curriculum utilizes five corpora per year, yielding approximately 315B tokens per yearly segment and summing to a total of 2.5T tokens. We generate checkpoints at the conclusion of each yearly segment, resulting in eight models corresponding to data cutoffs from 2018 to 2025. Consistent with the baseline, each chronological checkpoint is obtained following the 30,000-step cooldown phase described previously. This allows us to evaluate "fully converged" models at distinct temporal stages—e.g., the 2018 checkpoint acts as a model trained to convergence on 315B tokens, while the 2025 checkpoint covers the full 2.5T tokens. For comparative rigor, sequential checkpoints are evaluated against baseline checkpoints matched specifically by total token count. To ensure a fair comparison, we evaluate both models on knowledge up to 2024, as the shuffled baseline lacks more recent data. We extended the sequential training to 2025 for two reasons: to analyze more temporal checkpoints (five years once converged performance is reached) and to release the most up-to-date checkpoints to the community.

## 3. Evaluating Temporal Alignment

To evaluate the temporal alignment of our models, we construct a QA dataset centered on facts that evolve over time

*Table 1.* **Quantile distribution of subject popularity**. The shift in values from Wikidata to KairosQA demonstrate how our filtering prioritizes popular subjects to ensure robust temporal evaluation.

| Datasets | MIN | 25% | 50% | 75% | MAX |
|---|---|---|---|---|---|
| Wikidata | 1 | 340 | 1 378 | 6 151 | 133 423 662 |
| KairosQA | 161 | 27 277 | 78 194 | 320 176 | 44 800 923 |

as summarized in Fig. 2. We use Wikidata as our primary source due to its large scale, open availability, and explicit temporal annotations. From Wikidata, we extract subject–relation–object triplets associated with specific years, each serving as a single data sample.

**Filtering.** We restrict the dataset to a curated subset of Wikidata properties, or relations, that naturally exhibit temporal variation—specifically, relations whose associated answers change at least twice between 2018 and 2025. These properties concern people, organizations, sports, and events. To reduce noise from rare entities, we incorporate a popularity proxy metric based on Wikipedia page views. We prioritize popular subjects as we believe it serves as an indicator of the density of the information in our training set, ensuring that the evaluation measures temporal alignment rather than mere absence of knowledge. Starting from 17 million raw triplets, we apply successive filtering steps to ensure temporal validity, relevance, and sufficient temporal variation (Appendix C.1). We then select the top 20% most popular subjects (Tab. 1), a threshold that we found experimentally to provide a good trade-off between question difficulty and dataset coverage. The resulting dataset contains 7167 subject–relation pairs, each corresponding to a potential evaluation question. In Tab. 2, the number of available examples varies across years. For a given evaluation year, only pairs with valid answers for that year are retained, ensuring that models are always evaluated against accurate ground truth for that specific year. Once questions are generated, as discussed below, we further apply a relation-aware quality control step using Claude Sonnet[7] to detect and resolve relation-specific issues, including subjects leading to ambiguous or ill-defined questions, and incoherences between questions and their associated answers. Detected issues are resolved through Claude-assisted extraction from Wikipedia content, followed by a manual sanity check (Appendix C.2). The final relation distribution is dominated by sports and award-related facts, yielding a controlled yet diverse benchmark for assessing whether language models store and update temporally grounded knowledge rather than relying on static memorization.

**Generation.** We then generate diverse multiple-choice questions using GPT-4o mini via OpenRouter,[8] along with

---

[7] https://www.anthropic.com/claude
[8] https://openrouter.ai

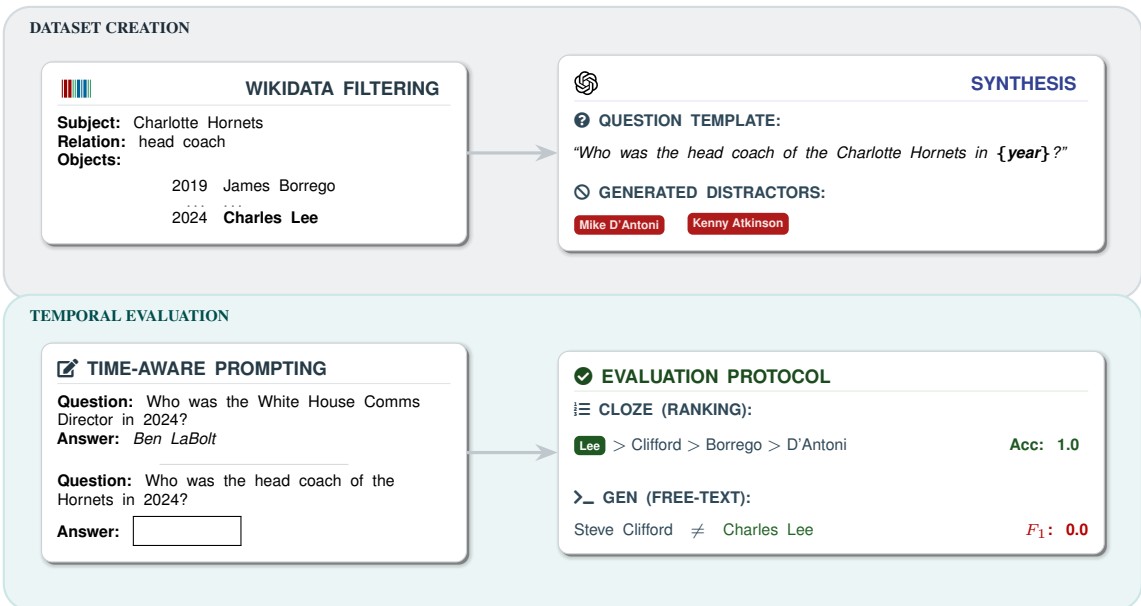

*Figure 2.* **Creation of KairosQA**. Summary of the methodology for creating the dataset of temporally sensitive facts (*top*) and the evaluation protocol (*bottom*) for the proposed benchmark of time-grounded knowledge.

*Table 2.* **Temporal distribution of evaluation examples.** Counts represent the subset of the 7 167 total subject–relation pairs that possess a ground-truth answer for each specific evaluation year.

| Years | 2014 | 2017 | 2020 | 2022 | 2024 | 2025 |
|---|---|---|---|---|---|---|
| Nb of Examples | 4 718 | 5 844 | 6 628 | 6 762 | 6 315 | 6 073 |

corresponding distractor answers. For each relation, we start from a reference template question and prompt the LLM to produce a modified version that should incorporate the target year, yielding a variety of coherent and contextually appropriate questions, as shown in Appendix C.4. Candidate answer choices are initially drawn from neighboring years around the target year, and if additional options are needed, we supplement them with distractors to reach the desired number of choices per question. At evaluation time, the target year is incorporated into each question. For certain target years, some questions have no valid answer and are therefore omitted. This process results in KairosQA, a temporally grounded question-answering dataset designed to evaluate the temporal reasoning capabilities of LLMs.

**Evaluation Protocol.** Following the OLMES benchmark (Gu et al., 2024), we adopt the cloze formulation (CF) (Brown et al., 2020). This approach is particularly effective for evaluating models that have not yet fully mastered the structural constraints of multiple-choice tasks. We observe this distinction clearly on MMLU (Hendrycks et al., 2021), where standard evaluation metrics (i.e., scoring the probability of option labels 'A', 'B', etc.) show a sudden discontinuity in accuracy—occurring around 200B tokens

for the shuffled baseline and 400B tokens for the sequential model, as shown in the Appendix 8. This jump correlates with the emergence of the model's ability to adhere to the multiple-choice format (MCF), which eventually yields higher accuracy than the CF. Because the OLMES protocol defines the score as the maximum of the two methods $(\max(\mathrm{MCF}, \mathrm{CF}))$, the final metric exhibits a sharp performance increase as it transitions to the superior MCF score. To mitigate this formatting bottleneck during early training, we therefore rely on the CF to capture latent knowledge.

To better reflect real-world usage, where models are queried without predefined answers, we complement the CF evaluation with a generative setting. We evaluate the generated responses using a normalized F1 score, following standard QA evaluation protocols (Rajpurkar et al., 2016). Specifically, we report the maximum score achieved across all valid answers for the target year. This combination enables us to study temporal preferences with minimal reliance on instruction-following ability, while still approximating realistic deployment scenarios. In the cloze setting, we uniformly sample the ground-truth answer from the valid answers for the target year. We then select the remaining multiple-choice options from neighboring years and a distractor list, ensuring that none of these distractors overlap with any valid answers for the target year. We normalize log-probabilities by the number of characters to reduce length bias, as this empirically proved most effective in our setup. Finally, although some questions in KairosQA remain ambiguous for open-ended generation despite extensive filtering, the use of constrained answer choices in the

cloze setting helps disambiguate the context and enables a more precise evaluation of temporally grounded knowledge.

## 4. Experimental settings

**Evaluation datasets.** On the one hand, we verify that sequential pre-training does not degrade model performance on downstream language modeling and general knowledge tasks. To this end, we rely on the OLMES benchmark (Gu et al., 2024), which covers a broad range of tasks and provides a precise, reproducible evaluation setup. On the other hand, we evaluate temporal factual knowledge using our benchmark KairosQA described in Section 3. To further strengthen our analysis, we additionally tested the models on TAQA (Zhao et al., 2024), a previously released time-sensitive dataset consisting of 9,000 questions extracted from Wikipedia tables covering 2000 to 2023.

**Our checkpoints.** As described in Section 2.3, we evaluate eight checkpoints for both shuffled and sequential pre-training, with pairwise matching token counts. For the sequential setup, checkpoints are taken after each yearly crawl from 2018 to 2025, corresponding to training budgets ranging from 315B to 2.5T tokens. Our primary focus is on comparing the sequential checkpoints from 2020 to 2024 with their shuffled counterparts, ensuring direct comparability while isolating the effect of temporal ordering. Throughout the rest of this paper, unless otherwise specified, the 2.5T token models are referred to as Shuffle and Sequential, according to their respective data ordering.

**Other open-source base models.** To validate our temporal evaluation dataset, we benchmark a range of open-source LLMs. We select both recently released models with parameter counts comparable to our own and earlier models to cover a broader range of training periods. When available, we report the official training cut-off dates; otherwise, we use the public release date as an upper bound on the training cut-off. The evaluated models include Llama 3.1-8B with a cut-off in December 2023 (Grattafiori et al., 2024), Gemma3 (4B) in August 2024 (Team et al., 2025), Olmo3 (7B) released in October 2025 (Olmo et al., 2026), and Qwen3 (4B, 8B and 14B) released in April 2025 (Yang et al., 2025).

## 5. Results

### 5.1. General Language Understanding

To validate our sequential training paradigm, we first assess whether chronological constraints hinder general domain convergence, relying on the OLMES benchmark (Gu et al., 2024). As illustrated in Fig. 3, the final performance of the sequential model is fully comparable to that of the baseline. This result validates that temporal ordering does not degrade general language understanding capabilities.

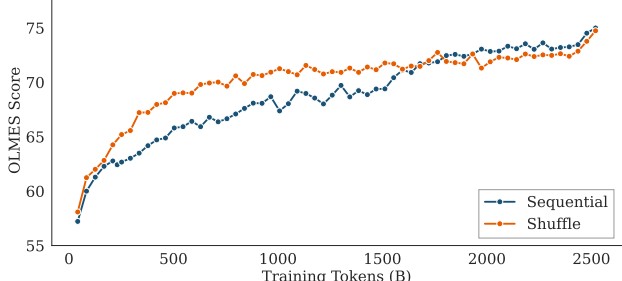

*Figure 3.* **Learning Dynamics on OLMES.** Evolution of general language understanding scores over 2.5T tokens. The Shuffle baseline exhibits higher efficiency in the mid-training phase, likely due to the stationary distribution of the data. The Sequential model steadily closes this gap, showing that chronological ordering alters the learning trajectory without compromising final model capacity.

However, the learning trajectories differ significantly between the two paradigms. The shuffle model maintains a consistent lead throughout half of the training, potentially benefiting from the stationary data distribution. In contrast, the sequential model exhibits a linear growth pattern, initially lagging behind the baseline. We hypothesize that this lag stems from the combined effects of sequential ordering constraints and non-stationary variations in data quality or density across different years. While our initial results in Section 5.5 provide preliminary evidence in support of this hypothesis, fully disentangling the specific impact of each factor remains a subject for future work. Despite this initial deficit, the performance gap proves to be transient: the sequential model closes the performance difference in the final third of training, eventually converging to parity with, and even surpassing, the baseline. This demonstrates that while temporal ordering alters the optimization path, it does not limit the model's final capacity.

### 5.2. Temporal analysis of the sequential pre-training

In this section, we provide an in-depth analysis of temporal dynamics across our checkpoints using the KairosQA benchmark. In Fig. 4, the cloze formulation allows us to assess the temporal alignment of models across temporally varying data distribution and different training budgets.

As illustrated in Fig. 9 (Appendix A.1), the shuffled checkpoints exhibit nearly identical performance dynamics across all pre-training lengths. This consistency indicates that grounding knowledge is not merely a function of data quantity; models trained on subsets of the data achieve roughly the same temporal alignment as those trained on the full corpus. Consequently, in Fig. 4, we report only the final shuffled checkpoint as a representative baseline. Notably, these baselines show a consistent degradation on recent temporal knowledge, with two distinct local maxima—in 2015 and 2020—followed by a precipitous drop toward ran-

**Temporal Evaluation Performance**

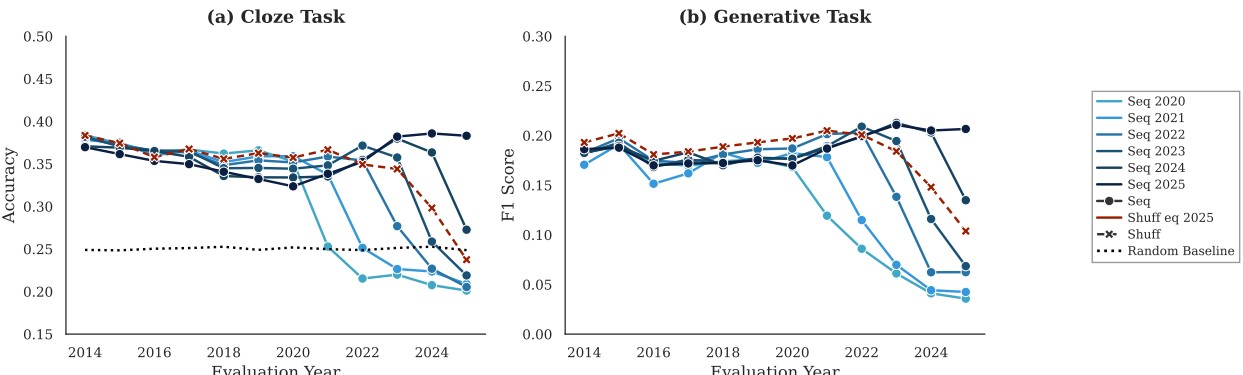

*Figure 4.* **Temporal evaluation on KairosQA.** Comparison of sequential checkpoints versus the last shuffled checkpoint, since shuffled checkpoints exhibit nearly identical performance dynamics across all pre-training lengths (see in Fig. 9). *Shuff eq* denotes the shuffled baseline trained on the same total token count as *Seq 2025*. (a) Cloze task accuracy across 4 choices; the random baseline varies slightly in cases where fewer than four choices were available. (b) Generative task performance measured by F1 score.

dom accuracy in 2024. This suggests that despite access to data spanning 2020–2024, the shuffled regime fails to effectively internalize contemporary knowledge, instead prioritizing historical information repeated across Common Crawl dumps. This observation reinforces the advantage of sequential training: since sufficient token volume exists per year to learn the concepts, the strict temporal ordering provides the necessary distributional shift to prioritize recent information, which simple volume scaling in the shuffled baseline fails to achieve.

In contrast, the sequential checkpoints display distinct behavioral patterns directly correlated with their training cut-off dates. While performance naturally declines for years following the training cutoff, each model consistently achieves its peak accuracy in the year immediately preceding its endpoint. This demonstrates that sequential training effectively targets and retains recent knowledge, unlike the shuffled baseline, successfully inducing a recency bias. Interestingly, the shuffled models exhibit performance levels comparable to the 2020–2021 sequential checkpoints, corresponding to the start of their training data period. This phenomenon suggests a form of *regression toward the mean*, aligning with similar observations of temporal alignment inertia reported in the TAQA study (Zhao et al., 2024).

A notable trade-off of this sequential method is the relative forgetting of older knowledge in favor of the new, as evidenced by the *Seq 2025* model in Fig. 4. These findings are further corroborated by the generative task results. In this setting, the increased training budget of the more recent checkpoints partially mitigates the forgetting effect; since later models have processed significantly more total tokens, their absolute generative performance remains high compared to earlier, less-trained checkpoints. Nevertheless, the

"recency peak" remains the dominant characteristic: sequential models excel in recent years where shuffled baselines experience their most significant failures.

**Other existing temporal evaluations.** To further assess temporal knowledge, we also evaluate models on the TAQA benchmark (Zhao et al., 2024). While TAQA is a well-established dataset for time-sensitive factual evaluation, we find that, in our setting, it provides limited discriminative power across the models considered. Under the standard protocol, without explicit year stated in the prompt and using time-insensitive in-context examples, models performance remain low, with F1 scores ranging from 1.5% to 10% and only marginal differences between models. Among the evaluated models, only Llama 3.1-8B surpasses an F1 score of 10% for the year 2019, see Appendix A.4 for further results. This behavior can be explained by differences in scale and scope between our experiments and prior work: TAQA evaluates temporal knowledge over facts ending in 2023 and was designed for large-scale models ($\geq$ 60B parameters), in contrast to our setting, which involves smaller architectures trained on more recent data. Although we observe improvements when applying explicit temporal prompting and time-sensitive in-context examples, the resulting performance gaps remain too small to draw strong conclusions. These results motivate the need for a complementary benchmark, better calibrated to our models and objectives, which we introduce to more precisely analyze temporal alignment and knowledge freshness.

### 5.3. Towards Temporal Freshness

Our analysis highlighted the advantages of training on sequentially ordered Common Crawl data over the standard practice of shuffling datasets regardless of their target year.

**Temporal Evaluation Performance**

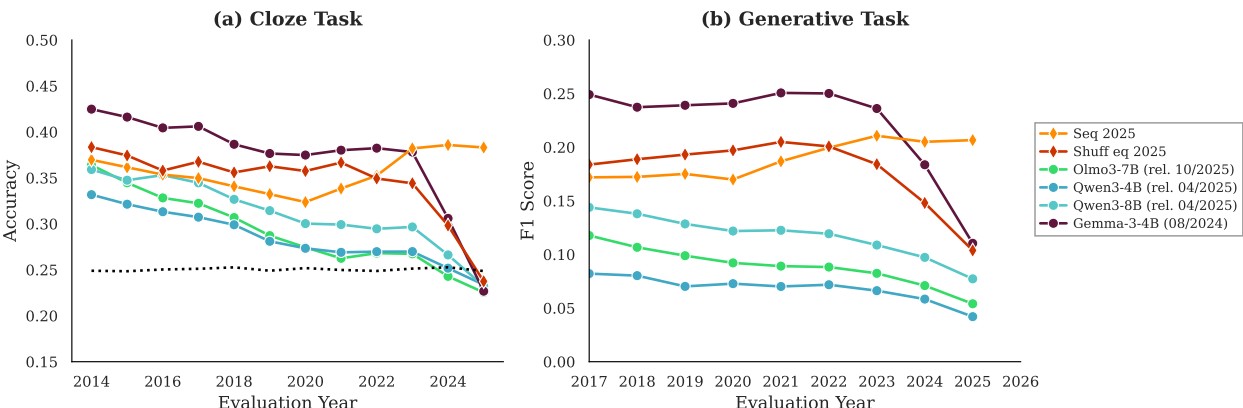

*Figure 5.* **Comparison with open-source models.** Temporal performance on KairosQA comparing our approach to various baseline models. (a) Displays accuracy in the cloze formulation, while (b) shows the corresponding F1 scores across evaluation years.

To further validate these findings, we demonstrate that most recently released open-source models exhibit temporal knowledge dynamics aligned with our shuffled baseline

As illustrated in Fig. 5, all evaluated open-source models exhibit a consistent performance degradation as questions become more recent. For recent data (2024), performance often approaches the random baseline. While the rate of decay varies, the trajectory remains uniform across the figures. Notably, model scale does not fundamentally alter this behavior (Appendix A.2); rather increased parameter counts merely shift the performance curve vertically within the same model family without fixing the temporal decay.

In contrast, as illustrated in Fig. 1, our sequential model is the only one to exhibit a reversal of this trend. It not only outperforms similarly sized open-source models on older knowledge, in Fig. 5, but also surpasses significantly larger models on recent knowledge (from 2023 onwards).

The generative task introduces additional complexity, as the model cannot rely on ranking candidates but must instead elicit specific knowledge from the precise temporal context. In this setting, while the performance curves for baseline models and the shuffled one are flatter, they still show a clear decline for recent years. For our sequential model, while the "forgetting" of older data is similarly flattened, the performance increase in recent years confirms the presence of more up-to-date, usable knowledge. Further comparisons with a broader range of open-source base models are available in Appendix A.2.

### 5.4. Other analyses of KairosQA

In this section, we further analyze our KairosQA benchmark. The following results are based on evaluations of

our last checkpoint from the sequential pre-training model. We focus on 2024 temporal knowledge as experimentally similar trends appear for the other targeted years and, in this study, we focus on fresh knowledge.

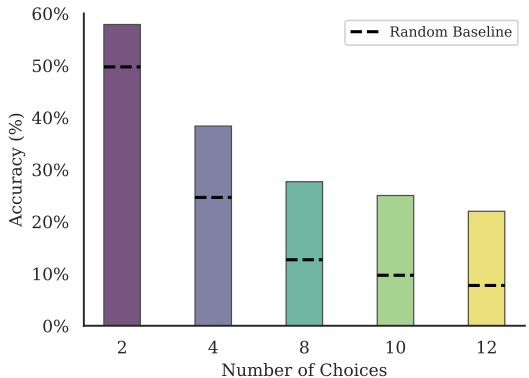

*Figure 6.* **Cloze formulation robustness.** We study the robustness of our protocol on our last checkpoint of the sequential pre-training by evaluating it on KairosQA for year 2024 in cloze formulation with an increasing number of choices.

**Cloze formulation robustness.** Cloze-style evaluation can be criticized as overly simplistic because it relies on ranking a finite set of candidates rather than on open-ended generation. Namely, if distractors are poorly selected, identifying the correct answer becomes trivial. However, Fig. 6 shows that the distractors in KairosQA are robust and effectively challenging. Even with only two choices, our model achieves approximately $58\%$ accuracy, which remains far from perfect but is still clearly above the random baseline. This gap is intentional: our distractors are valid answers for the same entity in different years, meaning both candidates are plausible unless the model possesses precise,

time-sensitive knowledge elicited by the prompt context.

Furthermore, if our model's performance relied solely on weak distractors, one would expect accuracy to collapse toward the random baseline as the number of choices increases. Instead, while performance naturally decreases, it stabilizes above 8 choices, maintaining a consistent margin of roughly 15 percentage points over the random baseline (e.g., reaching 22% accuracy for 12 choices against an 7.7% baseline). This stabilization indicates that the model is not merely guessing by elimination, but is instead leveraging a resilient internal representation of temporal facts that remains effective even as the task complexity increases.

Figure 7. **Effect of the popularity of the subject** in the questions on our last checkpoint of the sequential pre-training by evaluating it on KairosQA for year 2024.

**Subject popularity.** We also analyze question difficulty by categorizing subjects into popularity bins based on the proxy metric described in Section 3, which reflects subject frequency within the pre-training corpus. By targeting the most recent year (2024), we assess the model's ability to retrieve knowledge from a precise temporal context.

Fig. 7 illustrates a clear correlation between subject popularity and generative performance. The model demonstrates a strong mastery of "head" knowledge, achieving an F1 score of approximately 0.4 for the top 10% most popular subjects. However, a significant performance gap emerges in the "long tail"; performance drops sharply to an F1 of 0.17 for the 30–40% bin and plateaus between 0.10 and 0.15 for the bottom 50%. This suggests that while sequential pre-training captures up-to-date information, the "signal strength" required for accurate generations remains heavily dependent on training frequency. This underscores a persistent challenge: while the model outperforms larger shuffled baselines on recent knowledge, its mastery is non-uniform and remains weighted toward frequently mentioned entities.

### 5.5. Common Crawl snapshots quality across years

To explain the divergent learning trajectories of the shuffled and sequential models observed in Fig. 3, we hypothesize

that the performance lag stems mainly from non-stationary variations in data quality or density across different years.

To decouple the impact of data quality from the inherent cost of sequential learning, we conduct a targeted experiment: we perform the "cool-down" phase on the 2021 Sequential checkpoint using data from the 2024 snapshot instead of the original 2021 data. This intervention yields a +1.5% improvement on OLMES (73.2% vs. 71.7%) over the standard 2021-cool-down baseline. Because the training process remains fundamentally sequential, these results suggest that "sequential lag" is not an intrinsic flaw of the learning paradigm. Rather, they provide robust evidence for our hypothesis: the observed performance gap is primarily driven by variability in data quality across snapshots, at least between the 2021 and 2024 periods.

## 6. Related Work

**Pre-training data impact.** Since the seminal work of Brown et al. (2020), the pre-training dynamics of LLMs have been extensively studied. Prior work has shown that the composition and scheduling of training data can significantly influence downstream performance. For instance, Blakeney et al. (2024) demonstrate that upsampling domain-specific and code datasets toward the end of pre-training leads to improved performance on challenging tasks. Similarly, Feng et al. (2024) propose dividing pre-training data into two subsets with controlled domain distributions. Other lines of work directly investigate the relationship between training data and the factual knowledge encoded in model parameters. In controlled experimental settings, using synthetic data, Gu et al. (2026) identify phase transitions in knowledge acquisition that depend on both the proportion of observed facts and model scale. Zucchet et al. (2025) further analyze training dynamics related to factual learning and the emergence of hallucinations. In this work, we similarly investigate the impact of pre-training data curricula on knowledge acquisition. To maintain a realistic experimental setting, we avoid synthetic data in favor of production-scale pre-training techniques applied to real-world corpora. In parallel, recent efforts have contributed to democratizing LLM pre-training by releasing large-scale, filtered pre-training datasets. Recent examples include Penedo et al. (2024) and Maini et al. (2025), who construct high-quality corpora by filtering and synthesizing data from massive web crawls. Together, these works highlight the central role of data composition and training dynamics in shaping both the capabilities and knowledge of LLMs.

**Temporality in LLMs.** Subsequent work such as (Zhao et al., 2024) proposes a fine-tuning approach that enhances temporal awareness, while Park et al. (2025) introduce a prompting pipeline to target knowledge from specific years across multiple domains. We found the temporal bench-

marks proposed by both studies to be either too challenging or incompatible with our setting for measuring base model temporal alignment. Faro et al. (2025) propose a mixture-of-experts architecture with time-aware set of parameters to improve temporal accuracy. Most closely related to our work, Li et al. (2025) investigate continual learning strategies for incorporating more recent data by introducing month-specific corpora, exploring learning rate schedules, data replay mechanisms, and domain-specific data selection. In contrast to our study, which focuses on the role of pre-training dynamics and data ordering in shaping temporal knowledge, their work primarily addresses the challenge of retaining past knowledge under continual learning settings.

## 7. Perspectives & Conclusions

Our research demonstrates that the standard practice of shuffling pre-training data is a primary driver of the "knowledge horizon" gap observed in modern LLMs. By introducing KairosQA, a benchmark that isolates temporal awareness through rigorous filtering of Wikidata, we show that sequential pre-training allows models to achieve a "recency peak" that surpasses larger models on contemporary facts. While shuffled baselines suffer from temporal alignment inertia by over-prioritizing historical data repeated across web crawls, our sequential approach successfully induces the recency bias necessary for factual freshness.

However, the primary trade-off when using sequential pre-training is the relative forgetting of older knowledge as models adapt to newer temporal distributions. Although total token volume partially mitigates this, it highlights the need for future time-aware pre-training architectures that explicitly model the temporal origin of data to track evolving facts without erasing history. In Appendix A.3, we provide preliminary experiments that highlight the inherent difficulty of this challenge. By open-sourcing our checkpoints and dataset, we provide a foundation for research into continual learning and seamless incremental updates, demonstrating that temporal data ordering is a fundamental, under-explored factor in developing truly up-to-date base language models.

## Impact Statement

This paper presents work whose goal is to advance the field of Machine Learning. There are many potential societal consequences of our work, none which we feel must be specifically highlighted here.

## Acknowledgements.

This project is funded by Iliad Group, CMA CGM Group and Schmidt Sciences. We thank the Kyutai research team for their help and valuable feedback.

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

# A. Additional experiments

## A.1. More results on our checkpoints

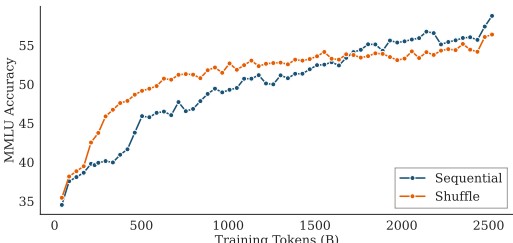

*Figure 8.* **Learning Dynamics on MMLU.** Evolution of MMLU scores over 2.5T tokens. The Shuffle baseline exhibits higher efficiency in the mid-training phase, likely due to the stationary distribution of the data. The Sequential model steadily closes this gap, showing that chronological ordering alters the learning trajectory without compromising final model capacity. The two discontinuities at 200B tokens for the Shuffle baseline and 400B for the other one illustrate the emergence of improved abilities on the multiple-choice task.

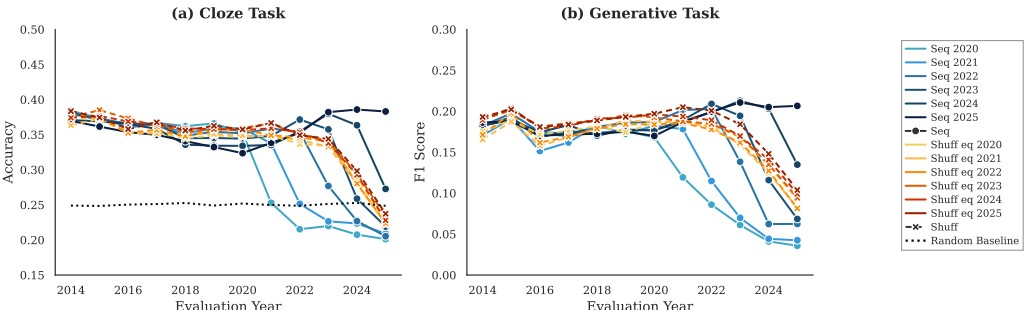

*Figure 9.* **Temporal evaluation on KairosQA.** Comparison of sequential checkpoints versus their shuffled counterparts in terms of total token count. Shuffled checkpoints exhibit nearly identical performance dynamics across all pre-training lengths. *Shuff eq 202\** denotes the shuffled baseline trained on the same total token count as *Seq 202\**. (a) Cloze task accuracy across 4 choices; the random baseline varies slightly in cases where fewer than four choices were available. (b) Generative task performance measured by F1 score.

## A.2. Open-source base models comparison

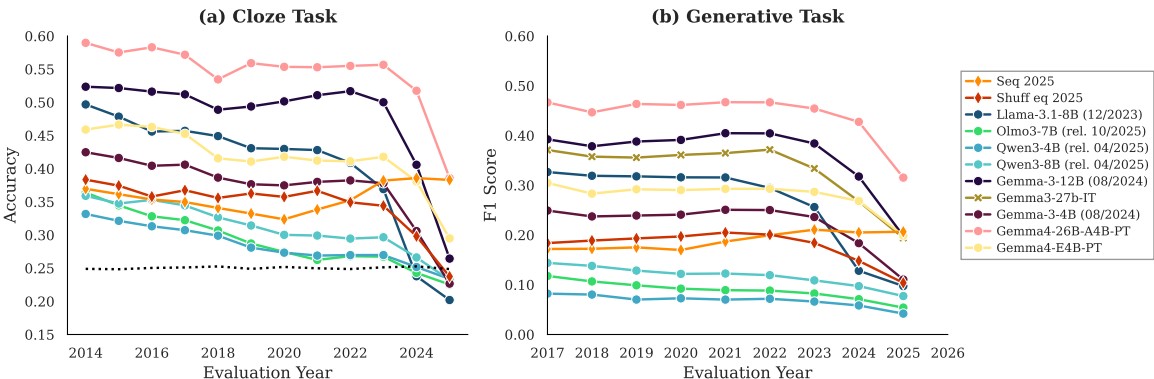

*Figure 10.* **Comparison with open-source models.** Temporal performance on KairosQA comparing our approach to various baseline models spanning different model sizes and cut-off dates. (a) Displays accuracy in the cloze formulation, while (b) shows the corresponding F1 scores across evaluation years.

## A.3. Mitigating the forgetting effect: a non-trivial challenge

Mitigating the loss of historical knowledge while sequentially training on contemporary data remains a critical challenge. Although our primary objective was to characterize this forgetting phenomenon, we explored several experimental directions to counteract it. However, these attempts proved inconclusive, highlighting the inherent difficulty of preserving past knowledge without compromising the integration of new information.

### A.3.1. MODEL SOUP

In Tab. 3, we present our experiments with model merging. Given that we have checkpoints for each year, we investigated whether a merged model could achieve superior performance across the entire timeline. To test this, we experimented with linear combinations of the weights from our "cooled-down" checkpoints. We began this process with the 2020 model, as the 2018 and 2019 checkpoints were under-trained and significantly degraded overall results.

Unfortunately, the merged model's performance appears to be an average of its components rather than a "best-of-all-worlds" maximum. Even with simple heuristics for our "model soup," the results show that incorporating earlier weights degrades performance on 2025 data while offering negligible recovery of older knowledge. These findings suggest that simple weight interpolation is insufficient to mitigate the catastrophic forgetting that occurs during sequential learning.

*Table 3.* Performance evaluation in cloze form across different soup coefficients for sequential merging (2018–2025).

| Model | Soup Coefficients | 2018 | 2019 | 2020 | 2021 | 2022 | 2023 | 2024 | 2025 |
|---|---|---|---|---|---|---|---|---|---|
| Seq 2025 | [0, 0, 0, 0, 0, 0, 0, 1] | **34.1** | **33.2** | **32.4** | **33.8** | **35.2** | **38.2** | **38.6** | **38.3** |
| Uniform | [1, 1, 1, 1, 1, 1, 1, 1] | 29.7 | 29.0 | 26.9 | 25.3 | 25.2 | 24.4 | 24.6 | 23.7 |
| Linear | [0.1, 0.2, 0.3, 0.4, 0.5, 0.6, 0.7, 0.8] | 30.2 | 30.1 | 29.6 | 28.2 | 28.7 | 28.7 | 25.7 | 23.3 |
| Exponential | [0.5, 1, 2, 4, 8, 16, 32, 64] | 32.3 | 32.3 | 31.7 | 32.2 | 33.8 | 35.2 | 33.5 | 29.6 |
| One Every Two | [0, 1, 0, 1, 0, 1, 0, 1] | 29.7 | 28.6 | 26.1 | 24.6 | 24.8 | 24.7 | 24.9 | 24.2 |

### A.3.2. REPLAYING FACTS FROM PREVIOUS YEARS

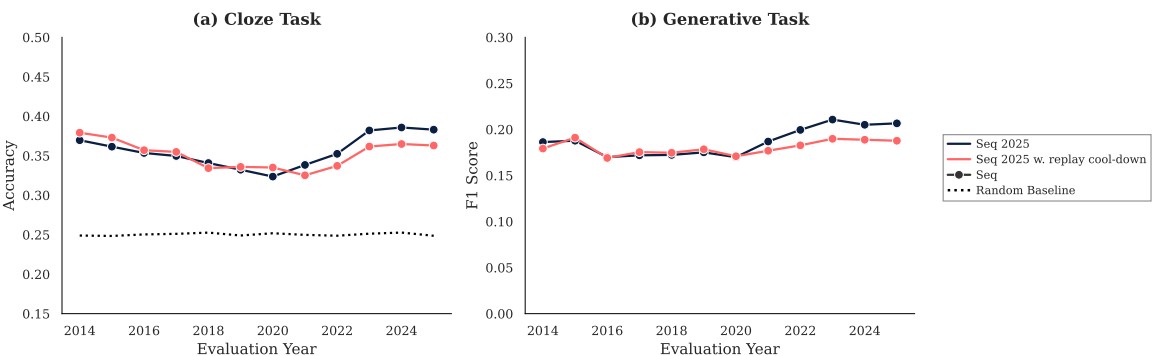

*Figure 11.* **Temporal evaluation on KairosQA.** Comparison of our last sequential checkpoint suffering from "relative forgetting" versus our version with replay of older knowledge as an attempt to mitigate forgetting.(a) Cloze task accuracy across 4 choices; the random baseline varies slightly in cases where fewer than four choices were available. (b) Generative task performance measured by F1 score.

To address forgetting in sequential training, we implement a "replay cool-down" during the final 30k training steps of the sequential model (Seq 2025). This process utilizes a 50/50 mixture of previously unseen 2025 crawl data and filtered 2020 crawl data, with a filtering strategy focused on educational content inspired by FineWeb (Penedo et al., 2024). As shown in Fig. 11, replaying the 2020 data slightly mitigates the loss of historical knowledge, though only on the cloze task, resulting in improved KairosQA performance compared to the standard sequential baseline. However, this gain comes at the cost of reduced performance on data from the 2021–2025 period. Ultimately, varying the replay ratio induces a zero-sum trade-off between preserving pre-2020 knowledge and sustaining modern performance, suggesting that simple replay is insufficient to fully resolve the forgetting inherent in sequential training.

### A.4. TAQA ([Zhao et al., 2024](#)) results

We implemented evaluations on the TAQA ([Zhao et al., 2024](#)) benchmark using the standard normalized F1 score. Our experiments covered both their baseline configuration and the time-aware prompting strategy, which aims to temporally re-align the LLM to target knowledge from a specific year.

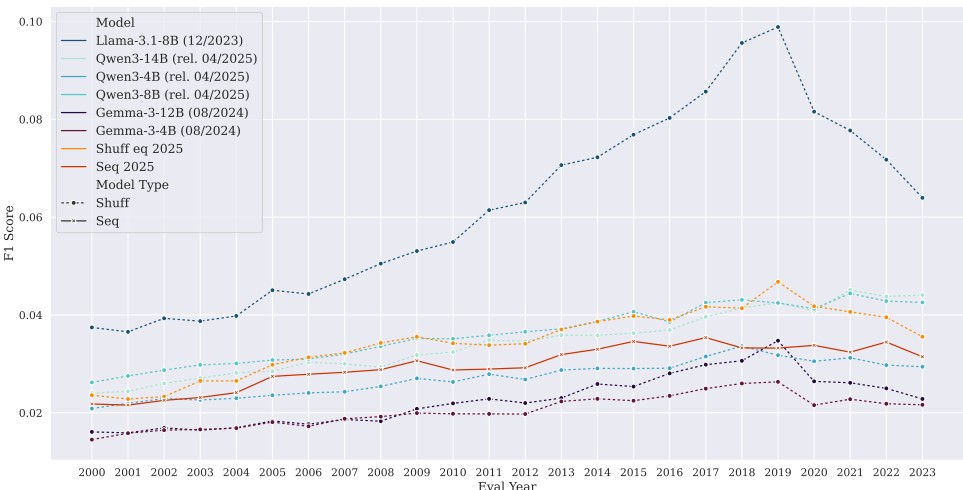

*Figure 12.* Evaluation of several open-source base models and our two pre-trained models on TAQA ([Zhao et al., 2024](#)) in the standard setting.

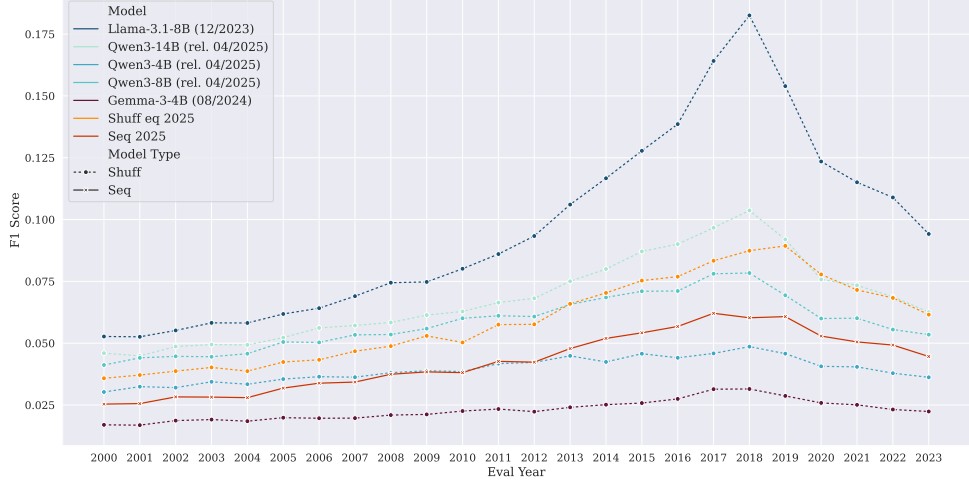

*Figure 13.* Evaluation of several open-source base models and our two pre-trained models on TAQA ([Zhao et al., 2024](#)) using the time-aware prompting strategy to target 2018.

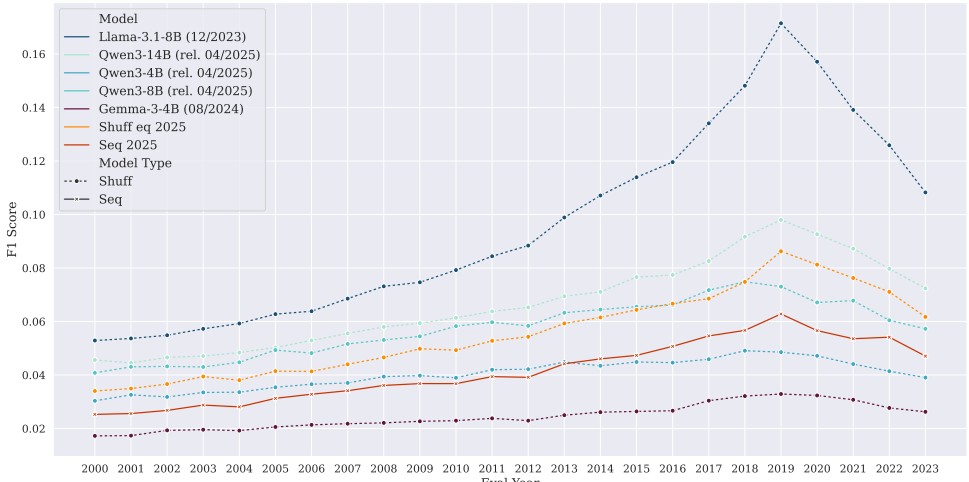

*Figure 14.* Evaluation of several open-source base models and our two pre-trained models on TAQA (Zhao et al., 2024) using the time-aware prompting strategy to target 2021.

## B. Pre-training setting

*Table 4.* Pre-training hyperparameters.

| Hyperparameters | pre-training settings |
|---|---|
| optimizer | AdamW |
| max lr | 1e-3 |
| lr scheduler type | Warmup-Stable-Decay |
| init lr | 0 |
| final lr | 1e-4 |
| warmup steps | 2000 |
| cooldown steps | 30000 |
| weight decay | 0.025 |
| batch size | 64 |
| effective batch size | $64 \times 16$ |
| GPUs | 128 H100 |
| max norm (gradient clipping) | 1.0 |
| flash attention | yes |
| number of tokens | 2.5T |

## C. Evaluation dataset

### C.1. Facts filtering

To construct our evaluation dataset, we first identify facts that evolve over time in order to assess the temporal alignment of our models. Wikidata is a well-suited source for this purpose, as it contains a large number of facts, over 120 million data items, while being open-source and maintaining a relatively high level of data quality. Each entry of Wikidata[9] consists in a dictionary with the following keys:

- *type*: string, indicates the type of entry (item vs. property)

- *id*: string, unique identifier in Wikidata

- *labels*: dictionary mapping languages to the name of the entry

- *descriptions*: dictionary mapping languages to a short description of the entry

- *aliases*: dictionary mapping languages to alternative names of the entry

- *claims*: dictionary containing the factual statements associated with the entry

- *sitelinks*: dictionary linking the entry to other Wikimedia projects (e.g., Wikipedia)

- *pageid*: integer, unique page identifier of the entry

- *ns*: integer, namespace identifier

- *title*: string containing the identifier of the entry

- *lastrevid*: integer, identifier of the most recent revision

- *modified*: timestamp indicating the date of the last revision

From this information, we construct triplets composed of a subject, a relation, and an object, each associated with a timestamp. Each such triplet constitutes a single data sample.

---

[9] https://doc.wikimedia.org/Wikibase/master/php/docs_topics_json.html

**Kept Wikidata properties.** We retain a subset of Wikidata properties corresponding to evolving factual information across people, organizations, sports, and events: **P54** (*member of sports team*); **P39** (*position held*, with qualifiers **P2389**, **P768**; subject/object inverted); **P286** (*head coach*); **P488** (*chairperson*); **P6** (*head of government*); **P35** (*head of state*); **P127** (*owned by*); **P1082** (*population*); **P166** (*award received*; inverted); **P26** (*spouse*); **P118** (*league or competition*); **P991** (*successful candidate*); **P1346** (*winner*, with qualifier **P2501**); **P822** (*mascot*); **P17** (*country*); **P276** (*location*); **P2882** (*relegated*); **P3279** (*standings*, with qualifiers **P1351**, **P1350**); **P108** (*employer*); **P102** (*member of political party*); **P69** (*educated at*); **P3342** (*significant event*); **P393** (*edition number*); **P10606** (*worn by*).

To add a "proxy" measure of how often the object appears in the training set we use `https://qrank.toolforge.org` to extract page views on Wikipedia. This enables us to select the most popular subjects to avoid using examples that are in the long-tail of the training set which would give a noisy signal.

Starting from 17M Wikidata triplets, we construct a subject-centric dictionary with the following structure, with $n$ the number of years when an object exists which changes across subject-relation pairs:

$$\text{subject} \rightarrow \begin{cases} \text{relation}_0 : \{\text{year}_0 \mapsto \text{object}, \dots, \text{year}_n \mapsto \text{object}\} \\ \text{relation}_1 : \{\dots\} \\ \vdots \end{cases}$$

We apply the following filtering steps:

1. **Temporal validity.** We remove triplets with missing start dates or unparseable timestamps, leaving **5.4M subjects**.

2. **Population-only subjects.** Since population requires exact numerical answers and introduces excessive complexity, we remove subjects whose only relation is *population*, resulting in **4.9M subjects**.

3. **Popularity constraint.** We discard subjects without a popularity metric (`subject_rank`), leaving **1.9M subjects**.

4. **Relation distribution.** At this stage, 96 relations remain. The top relations include *educated at* (30.1%), *employer* (26.4%), *spouse* (11.9%), and *member of sports team* (11.7%).

5. **Temporal variation constraint.** We retain only subject–relation pairs whose answers change at least twice between 2018 and 2025, reducing the dataset to **35,246 subjects** and 48 relations.

6. **Popularity-based pruning.** Selecting the top 20% most popular subjects (`subject_rank` > 11k) yields **7,050 subjects** and approximately **7,400 subject–relation pairs**.

7. **Final pruning.** After removing the *population* relation, we obtain **7,268 subject–relation pairs**, corresponding to as many potential evaluation questions. For a given evaluation year (e.g., 2018), only pairs with valid answers for that year are retained (e.g., 6,200 pairs for 2018). The final relations proportion are highlighted in Table. 5.

### C.2. Relation-Aware Filtering.

Beyond the general filtering steps described above, we apply a relation-specific quality control pipeline assisted by Claude.[10] We detail below the cleaning procedure applied to the *award received* relation, where we identified the most critical issues.

**Achiever–work ambiguity.** For a given award, Wikidata sometimes conflates two semantically distinct answer types: the *achiever* (e.g., a singer or a film director) and the *work* (e.g., a song or an album). Although this does not affect the multiple-choice setting—where the correct answer is always provided—it does impact the generative evaluation, since a model may produce either type as a valid response. To resolve this, we split affected awards into two separate question variants: one targeting the achiever (e.g., *"Who received the Album of the Year award in {year}?"*) and one targeting the work (e.g., *"Which album received the Album of the Year award in {year}?"*). Claude is used to determine which awards require this split. As splitting may leave certain years without a valid answer, we use Claude alongside the award's Wikipedia page to extract missing winners and to verify existing answers. Any answer extracted by Claude that differs from the one originally provided in the dataset is flagged and resolved through manual review.

---

[10] `https://www.anthropic.com/claude`

**Ill-defined awards.** Claude is also used to flag awards that are structurally unsuitable for evaluation. Three categories of issues are identified:

- *Ambiguous scope*: awards spanning multiple subcategories with no further specification yield ill-formed questions (e.g., *"Who received the Grammy Award in {year}?"*). For such cases, Claude identifies the most prominent subcategories, and we retrieve the corresponding Wikipedia pages, which depending on the award can be: a single page listing all winners across years and subcategories, a per-year page listing all subcategory winners for that year, or a per-subcategory page listing all winners across years. The retrieved content is then fed to Claude to extract the corresponding winners, followed by a manual sanity check.

- *Multi-winner awards*: awards granted simultaneously to several recipients with no means of disambiguation (e.g., medals with multiple recipients per year), where complete ground-truth coverage in Wikidata cannot be guaranteed. Such awards are removed from the dataset.

- *Other ill-defined awards*: any award deemed unsuitable for unambiguous evaluation for other reasons. Such awards are likewise removed from the dataset.

**Answer coherence.** Finally, Claude is used to detect incoherences between questions and their associated answers, which typically arise from erroneous Wikidata extractions. Observed cases include, among others: a coach answer matched to the wrong-gender team for the *head coach* relation, a runner-up listed in place of the actual winner for the *winner* relation, a prominent organizational figure incorrectly assigned to the *chairperson* relation, or answers conflating two individuals who share the same name but have distinct careers for career-oriented relations such as *member of a sport team*. All flagged cases are manually verified and removed if necessary.

*Table 5.* Proportion of relations by year (2014–2025)

| Relation Name | 2014 | 2015 | 2016 | 2017 | 2018 | 2019 | 2020 | 2021 | 2022 | 2023 | 2024 | 2025 |
|---|---|---|---|---|---|---|---|---|---|---|---|---|
| Member of sports team | 38.3 | 40.2 | 41.1 | 42.4 | 42.1 | 42.5 | 42.7 | 42.0 | 40.8 | 39.7 | 38.3 | 38.6 |
| Award received | 26.7 | 25.1 | 24.0 | 22.8 | 21.5 | 20.8 | 19.9 | 19.8 | 20.6 | 21.2 | 21.7 | 20.5 |
| Position held | 15.1 | 14.9 | 14.2 | 13.6 | 13.0 | 12.7 | 13.0 | 13.3 | 13.3 | 13.5 | 13.4 | 13.4 |
| Head coach | 7.0 | 7.3 | 8.4 | 9.3 | 11.9 | 12.7 | 13.6 | 13.5 | 13.8 | 13.7 | 14.7 | 15.3 |
| Winner | 3.2 | 3.1 | 2.9 | 2.8 | 2.6 | 2.5 | 1.9 | 2.5 | 2.7 | 2.8 | 2.8 | 2.9 |
| Chairperson | 3.4 | 3.3 | 3.1 | 3.0 | 2.9 | 2.8 | 2.9 | 2.8 | 2.8 | 2.9 | 2.9 | 3.0 |
| Employer | 1.2 | 1.1 | 1.2 | 1.2 | 1.2 | 1.1 | 1.1 | 1.1 | 1.1 | 1.1 | 1.1 | 1.1 |
| Member of political party | 0.9 | 0.9 | 0.9 | 0.9 | 0.9 | 0.9 | 0.9 | 0.9 | 0.8 | 0.9 | 0.9 | 0.9 |
| League or competition | 0.6 | 0.6 | 0.7 | 0.7 | 0.8 | 0.8 | 0.9 | 1.0 | 1.0 | 1.0 | 1.0 | 1.0 |
| **Nb of Examples** | 4718 | 5054 | 5449 | 5844 | 6365 | 6630 | 6628 | 6762 | 6762 | 6535 | 6315 | 6073 |

## C.3. KairosQA generation templates

Experimental results indicate that ambiguous formulations of subjects or relations frequently result in imprecise generated questions. While these generation errors are more prevalent within specific subsets of relations, they can also occur sporadically across other relations depending on the subject context. To mitigate this, we determined that the most robust approach is to manually predefine a specific question template for each relation. This ensures the question remains structurally sound and targets the correct answer. Subsequently, we leverage GPT-4o mini to reformulate the question for the specific relation, providing the model with access to the specific subject to maintain contextual accuracy. To ensure the dataset supports a diverse distribution of question types and is specifically optimized for temporal evaluation, we instruct the LLM to include a designated placeholder for the target year. This structural design ensures that each reformulated question is intrinsically "time-aware", allowing for the precise elicitation of knowledge associated with a specific point in time.

---

**Question template**

You are an expert in natural language processing. Your task is to generate a clear and concise question about what or who is associated with subject "`<subject>`" through the relation "`<relation>`".
**Example:**
`<question>`
**Requirements:**

- Be creative to not use exactly the same phrasing each time.

- The question should be suited for multiple years, adding a {{year}} indicator for the user to change it.

**Output format:**
Question: `<your generated question here>`.

---

**PH Question Template**

You are an expert in natural language processing. Your task is to reformulate if necessary the following question making sure that "`<subject>`" still appears: "In {{year}}, who held the position of `<subject>`?".
**Output Format:**
Question: `<your generated question here>`.

---

**AR Question Template**

You are an expert in natural language processing. Your task is to reformulate if necessary the following question making sure that "`<subject>`" still appears: "In {{year}}, who received the `<subject>`?".
**Output Format:**
Question: `<your generated question here>`.

| Predefined Question Templates | |
|---|---|
| **Subject Type** | **Question Template** |
| head of government | Who was the head of government of `<subject>` in {{year}}? |
| head of state | Who was the head of state of `<subject>` in {{year}}? |
| position held | Who held the position of `<subject>` in {{year}}? |
| award received | Who received the `<subject>` in {{year}}? |
| winner | What did `<subject>` win in {{year}}? |
| member of sports team | Which sports team did `<subject>` play for in {{year}}? |
| head coach | Who was the head coach of `<subject>` in {{year}}? |
| league or competition | In which league or competition did `<subject>` compete in {{year}}? |
| chairperson | Who served as the chairperson of `<subject>` in {{year}}? |
| overall winner general classification | Who was the overall winner of the general classification for the `<subject>` in {{year}}? |
| second overall | Who finished second overall in the `<subject>` in {{year}}? |
| third overall | Who finished third overall in the `<subject>` in {{year}}? |
| winner of the points classification | Who won the points classification in the `<subject>` in {{year}}? |
| winner of the mountain classification | Who won the mountain classification in the `<subject>` in {{year}}? |
| winner of the young rider classification | Who won the young rider classification in the `<subject>` in {{year}}? |
| winner of the teams classification by time | Which team won the teams classification by time in the `<subject>` in {{year}}? |
| winner of the most combative rider | Who was named the most combative rider in the `<subject>` in {{year}}? |
| employer | Who employed `<subject>` in {{year}}? |
| member of political party | Which political party was `<subject>` affiliated with in {{year}}? |
| educated at | Where did `<subject>` receive their education in {{year}}? |
| owned by | Who owned `<subject>` in {{year}}? |
| stage winner | Who won the stage in the `<subject>` in {{year}}? |
| leader of the young rider classification | Who was leading the young rider classification in the `<subject>` in {{year}}? |
| leader of the points classification | Who was leading the points classification in the `<subject>` in {{year}}? |
| leader of the mountain classification | Who was leading the mountain classification in the `<subject>` in {{year}}? |
| leader of the teams classification by time | Which team was leading the teams classification by time in the `<subject>` in {{year}}? |
| most combative rider | Who was considered the most combative rider in the `<subject>` in {{year}}? |
| overall leader at the end of the stage | Who was the overall leader at the end of the stage in the `<subject>` in {{year}}? |
| winner of the sprint classification | Who won the sprint classification in the `<subject>` in {{year}}? |

To generate high-quality distractors, we provide the model with access to the full set of valid answers for a given subject across all recorded years (post-2020). We then apply a rigorous filtering process to ensure the generated distractors are distinct from one another and from the ground-truth answers. This deduplication is achieved through a hybrid approach combining F1-score thresholds and fuzzy matching metrics to eliminate redundant candidates.

---

**Distractor Template**

You are an expert in natural language processing and logic puzzles, skilled at generating plausible yet misleading distractor options that challenge users to distinguish between correct and incorrect answers. Your task is to create a distractor answer for the following question that is plausible but incorrect, ensuring it does not match the correct answer provided.

**Input:**
Question: `<question>`
Existing Answers: `<answer>`
**Requirements:**

- The distractor must be plausible and relevant to the question.

- It should not be the same as the existing answer.

- Ensure that the distractor is distinct from the existing answer.

**Output Format:**
Distractor: `<your distractor answer here>`.

### C.4. KairosQA examples

> **Temporal Question Example**
>
> **Prompt Input (5-Shot Context)**
> The following questions have been answered.
>
> Question: Who was awarded the Nishan-e-Pakistan in 2024?
> Answer: Rahim Aga Khan
>
> Question: Who was the Prefect of French Guiana in 2024?
> Answer: Antoine Poussier
>
> Question: Which sports team was Bilal El Khannouss a part of in 2024?
> Answer: Leicester City F.C.
>
> Question: Who held the position of head coach for Willem II during the year 2024?
> Answer: Peter Maes
>
> Question: Which sports team was Rafa Mir a member of during the year 2024?
> Answer: Sevilla FC
>
> Question: Who was awarded the insect of the year in 2024?
> Answer:
>
> ---
>
> **Multiple Choice Options:**
>
> 1. `[2025]` Rhyssa persuasoria
>
> 2. `[2023]` Araschnia levana
>
> 3. `[2022]` Venustoraphidia nigricollis
>
> 4. `[2024]` **Typhaeus typhoeus**
>
> 5. `[Distractor]` Cerambyx cerdo

