# OpenReview forum: "Understanding Data Temporality Impact on Large Language Models Pre-training"
_ICML.cc/2026/Conference — ICML 2026 regular_

### Official Review · Reviewer_hyVT · 2026-03-08

**Soundness:** 3
**Presentation:** 3
**Significance:** 3
**Originality:** 3
**Overall Recommendation:** 5
**Confidence:** 4

**Summary:**

Large Language models are pre-trained on shuffled data. The authors hypothesise that this shuffling could have an impact on models’ ability to capture temporal information.

To this end they:
a) create a dataset consisting of 7k temporally grounded questions from wikipedia where the associated facts have changed; and
b) they pre-train a 6B model with temporally ordered data. The results indicate that the temporally ordered pre-training regime does not affect the general knowledge understanding negatively, but does improve temporal information encoded within the models.

**Compliance With Llm Reviewing Policy:**

Affirmed.

**Final Justification:**

The detailed rebuttal and additional experiments addressed my main concerns and I have changed my evaluation.

It also reinforced my prior assessment regarding forgetting being an issue, but I also see that as future work, and not necessarily something that has to be solved as part of this work.

**Key Questions For Authors:**

Can you evaluate a larger model using your dataset to see if increased parameters offset the limitation demonstrated?

Can you evaluate a model that explicitly has a knowledge-cutoff after the last date in your evaluation dataset (2024)? This will ensure that the this is not a confound?

Up to 2019, the performance of the model trained on temporal snapshots drops - this is understandable due to forgetting of old data. Is it possible to add some high frequency data/facts from previous years to mitigate this?

**Limitations:**

Yes

**Strengths And Weaknesses:**

Soundness:
The experimental setup is overall sound and presents a mechanism to provide temporal grounding during pre-training. The performance on the temporal evaluation of the temporally grounded pre-training is large.

The topics associated with various topics are not always the same across different years. Therefore, it is possible that this might be a confounding factor. While the performance difference is so large that this is largely not a significant factor, it will be nice to see an evaluation on a topic balanced subset of the dataset.

Since pre-training additional models is expensive, it will be nice to see stronger external validation. Specifically: a) expand the evaluation to larger models to see if they are able to better capture more recent information from pre-training, and b) evaluate on a model for which the knowledge cut-off is explicitly after the test window (instead of using release date which might be well after knowledge cutoff. (See questions)


Presentation:
The paper is clearly written and the narrative is clear. The results are clearly presented and easy to understand.


Significance:
This method is significant for training LLMs and also for understanding how LLMs capture knowledge from pre-training data.


Originality:
This work is different from prior work in how it goes about creating the evaluation dataset and setup of pre-training and is therefore original.

---

> ### Author Rebuttal · Authors · 2026-03-31
>
> We thank the reviewer for recognizing the significance of our findings and the clarity of our narrative. We appreciate their constructive proposals and have addressed them in the following responses.
>
> ---
>
> &nbsp;
>
> ### Q1/Q2. Expand the evaluation to larger models and explicit knowledge cut-offs
>
> We address both sub-points (1) and (2) together, as our extended evaluations encompass both concerns. Our response relies on a combination of new experimental results (in the table below) and existing data from Appendix A (Fig. 10), which further details the phenomena observed in the main text.
> - **(1) Larger models.** While absolute scores vary across model families, a consistent decline in performance in more recent years persists across all scales.
> - **(2) Explicit knowledge cut-offs.** We evaluate several models with documented cut-offs:
>     - Llama 3.1-8B (December 2023) [1]
>     - Gemma 3 (August 2024) [2]
>     - OLMo 3, whose official cut-off is December 2024 despite their October 2025 release date [3].
>
>     All exhibit the same degradation pattern as our shuffle baseline with performance declining approximately one year before the stated cut-off.
>
> _Table: Results on KairosQA (generative task) for larger models_
> | Model name | 2018 | 2019 | 2020 | 2021 | 2022 | 2023 | 2024 | 2025 |
> | :--- | :---: | :---: | :---: | :---: | :---: | :---: | :---: | :---: |
> | **Seq 2025** | 14.3 | 13.3 | 14.5 | 14.9 | 15.7 | 16.2 | 17.7 | 19.2 |
> | **Shuff 2025** | 14.4 | 15.1 | 17.1 | 17.2 | 16.3 | 15.3 | 13.1 | 10.3 |
> | **Olmo3-32B-IT** | 11.2 | 10.1 | 10.1 | 9.6 | 9.5 | 9.6 | 8.2 | 6.5 |
> | **Gemma3-27B-IT** | 27.3 | 27.1 | 29.3 | 28.3 | 27.7 | 26.9 | 22.0 | 17.5 |
>
> &nbsp;
>
> ### Q3. Mitigating Forgetting
>
> We agree that mitigating the forgetting of older knowledge during sequential training on fresher facts would be a crucial improvement. Therefore, as proposed by the reviewer, we performed a ‘replay cool-down’ over the final 30k training steps of the sequential model (Seq 2025), using a 50/50 mixture of remaining unseen data from the 2025 crawl and filtered educational data from the 2020 crawl (inspired by FineWeb [4]).
>
> &nbsp;
>
> _Table: Results on KairosQA (generative task) for our model cool-downed with replay_
>  | Model name | 2014 | 2015 | 2016 | 2017 | 2018 | 2019 | 2020 | 2021 | 2022 | 2023 | 2024 | 2025 |
> | :--- | :---: | :---: | :---: | :---: | :---: | :---: | :---: | :---: | :---: | :---: | :---: | :---: |
> | **Seq 2025** | 15.1 | 15.4 | 15.9 | 15.6 | **14.3**  | 13.3 | 14.5  | **14.9**  | **15.7**  | **16.2**  | **17.7**  | **19.2**  |
> | **Seq 2025 w. 2020 replay** | **18.3**  | **18.1**  | **17.2**  | **17.2**  | 13.9 | **13.8**  | **14.6** | 13.6 | 14.1 | 15.2 | 15.8 | 16.2 |
>
> As shown in the above table, replaying 2020 data successfully improves retention of pre-2020 knowledge but comes at the expense of reduced performance over 2021-2025. This reveals the recency-rentention trade-off: replay covers older facts but dilutes the recent ones, decreasing the sequential model’s advantage.
>
> We also explored model merging (see our response to 8jn7, Q1), which similarly failed to recover older knowledge without degrading recent performance. Together, these results suggest that simple post-hoc interventions, whether weight interpolation or data replay, are insufficient to resolve forgetting, motivating more thorough investigations as future work.
>
> &nbsp;
>
> ---
>
> [1] Aaron Grattafiori et al, "The Llama 3 Herd of Models," 2024.
>
> [2] Gemma Team, et al, "Gemma 3 Technical Report," 2025.
>
> [3] Team Olmo, et al, "Olmo 3," 2025.
>
> [4] Guilherme Penedo, et al, "The FineWeb Datasets: Decanting the Web for the Finest Text Data at Scale," 2024.

---

> > ### Author Rebuttal · Reviewer_hyVT · 2026-04-03
> >
> > Thank you for the detailed response and the additional experiments.
> >
> > The analysis of the cut-off dates strengthens the paper, thank you.
> >
> > I understand that forgetting is an issue, but I also see that as future work, and not necessarily something that has to be solved as part of this work.
> >
> > I have increased my score.

---

### Official Review · Reviewer_h6u5 · 2026-03-12

**Soundness:** 2
**Presentation:** 3
**Significance:** 3
**Originality:** 3
**Overall Recommendation:** 4
**Confidence:** 4

**Summary:**

The paper compares models that have been sequentially trained vs trained with shuffled temporal training data. It introduces a new benchmark to support its findings, and shows that sequential training performs better at recalling fresh information and matches general performance compared to a model trained on randomly sorted temporal training data.

**Compliance With Llm Reviewing Policy:**

Affirmed.

**Key Questions For Authors:**

1) Have you done some tests on whether the temporal granularity of the ordering affects results?
2) Was there any human validation of the generated questions in KairosQA Appendix C.2?
3) The sequentially trained model is trained on data from 2018 to 2025, whereas the shuffled model was trained on data from 2020 to 2024. Could you train the models on the same data? This would better isolate the effect of data sequencing.

**Limitations:**

- The asymmetry of the temporal coverage of the training data.
- The forgetting phenomenon is not fully addressed.

**Strengths And Weaknesses:**

Strengths:
- The research question is interesting and well-motivated.
- Good breadth of evaluation:
   - OLMES, KairosQA, and TAQA
   - Several models (Llama, Gemma, Qwen, OLMo)
- Clear figures and tables that support the message of the paper well. Figure 4 is a really well-constructed figure that clearly encapsulates the point of the paper.
- A new dataset that will be useful to other researchers. The problem is hard, with the models far from benchmark saturation. The robustness test in Figure 6 is also pretty great, and shows that the benchmark is capturing the true signal rather than noise and process of elimination.


Weaknesses:
- The paper only tested one model. Hoever given the extensive compute needed for training, this is permissible.
- The temporal knowledge in the shuffled and sequential models is not the same. Sequential: 2018-2025. Shuffled: 2020-2024. It is a little odd why they chose to do this; it weakens their claims.
- Kairos is dominated by two domains: sports and politics. There is no attempt to address this, but it is an artefact of WikiData.

---

> ### Author Rebuttal · Authors · 2026-03-31
>
> We thank the reviewer for recognizing the breadth of our evaluation and the robustness of KairosQA. We appreciate the positive feedback and hope our detailed responses further demonstrate the rigor of our experimental protocol.
>
> ---
>
> &nbsp;
>
> ### W1. Asymmetric temporal coverage.
>
> Our shuffled baseline was trained before we started the project, and reflects standard practice of training on a few years of Common Crawl (CC) snapshots, consistent with how most open-source LLMs are pre-trained (eg. Llama 3, OLMo 3). For the sequential model, we decided to include data from a wider time period, to obtain checkpoints capturing more temporal facts than just from 2020-2024, as we believed this would enable more studies of temporal knowledge in LLM pre-training. We do not believe that including data from 2018-2019 would give an unfair advantage to the sequential model: data from older snapshots tend to yield models of lower quality, as observed in FineWeb [1]. We confirmed this observation by taking the 2021 snapshot and performed a cooldown on data from 2024 instead of 2021, leading to a 1.5% improvement on OLMES. Including data from 2025 does not impact our results, as our observations hold on the model up-to 2024, but it provides an additional temporal checkpoint for analysis and allows the release of more up-to-date models to the community.
>
> Regarding the reviewer comment on training both models on the same temporal range, it is a fair suggestion and we agree it would be a valuable ablation if compute resources permit it. That said, we want to emphasize that the paper's primary contribution is not to advocate sequential pre-training as a method, but rather to analyze how data ordering shapes temporal knowledge dynamics and to provide the community with KairosQA as a tool for measuring this.
>
> &nbsp;
>
> ### W2. KairosQA is dominated by sports and politics.
>
> This is a fair observation. The concentration stems from both Wikidata's coverage and our deliberate choice to prioritize popular subjects as a proxy for training set density (Section 3). Sports and career facts are particularly suited to temporal evaluation because they change frequently and predictably, providing a clean signal. Importantly, Table 4 shows the domain distribution is stable across years, so temporal trends in our results are not confounded by shifting domain composition.
>
> &nbsp;
>
> ###  Q1. Temporal granularity of the ordering affects results
>
> Our sequential training operates at the granularity of individual CC snapshots (approximately five per year), not at a yearly level. We therefore have sub-yearly checkpoints available. However, for evaluation purposes, we constructed KairosQA at yearly granularity because most Wikidata facts are annotated with year-level timestamps, making finer-grained temporal evaluation impractical with this data source.
>
> &nbsp;
>
> ###  Q2. Human validation of the generated questions
>
>
> We conducted multiple rounds of manual inspection by randomly sampling generated questions across all relation types. This iterative process allowed us to identify systematic failure modes in the question generation (such as ambiguous formulations for specific relation types) and refine the prompts accordingly over several iterations. This is what led us to adopt the relation specific template approach described in Appendix C.2, which proved substantially more robust than a single generic prompt.
>
> &nbsp;
> ### Limitations
> We agree that temporal forgetting requires deeper investigation. We explored preliminary mitigation strategies including: model merging (see our response to 8jn7, Q1), and a replay cool-down phase including data from previous years (see our response to hyVT, Q3). These initial attempts proved inconclusive, highlighting the non-trivial nature of the task and motivating a more in-depth study as future work.

---

### Official Review · Reviewer_8jn7 · 2026-03-13

**Soundness:** 3
**Presentation:** 3
**Significance:** 3
**Originality:** 3
**Overall Recommendation:** 4
**Confidence:** 3

**Summary:**

This paper investigates how the chronological order of pre-training data affects the acquisition and retention of time-sensitive factual knowledge in Large Language Models (LLMs). The authors challenge the standard practice of training on shuffled corpora, which they argue leads to a "knowledge horizon" gap where models struggle with facts near their training cutoff. To study this, they pre-train 6B-parameter models on 2.5T tokens using two regimes: one with globally shuffled data and another where Common Crawl snapshots are processed in strict chronological order from 2018 to 2025.

**Compliance With Llm Reviewing Policy:**

Affirmed.

**Key Questions For Authors:**

See weakness.

**Limitations:**

yes

**Strengths And Weaknesses:**

Strength:
1. The experimental design is exceptionally sound. By keeping the architecture, hyperparameters, and total token count (2.5T) identical between the "Shuffle" and "Sequential" models, the authors successfully isolate data ordering as the independent variable.
2. The use of a "branching cooldown" strategy ensures that comparisons are made between fully converged models at each stage, rather than mid-optimization artifacts.
3. The methodology for creating KairosQA is meticulous.  The authors didn't just extract triplets;  they filtered for "popularity" using Wikipedia page views to ensure they were testing "head" knowledge rather than "long-tail" noise.  Furthermore, using both Cloze (ranking) and Generative (F1) tasks allows for a robust assessment that separates internal knowledge from a model's ability to follow multiple-choice formatting.
4. The paper provides a fresh perspective on why LLMs struggle with recent facts. While prior work focused on fine-tuning or RAG, this work demonstrates that the issue begins at the pre-training stage due to data shuffling. The discovery that shuffled models prioritize older, repeated historical data over newer information is a key insight.

Weakness:
1.	The results clearly show that sequential models suffer from "relative forgetting" of older knowledge as they adapt to newer distributions. The paper identifies this trade-off but provides little exploration of how to mitigate it within the pre-training phase.
2.	Figure shows the Shuffle baseline is more efficient in the mid-training phase, while the Sequential model lags before eventually catching up. The authors hypothesize this is due to "non-stationary variations in data quality" but do not provide evidence. It is unclear if the lag is an inherent cost of sequential learning or just a byproduct of the specific data in the 2018-2021 snapshots.
3.	The authors note that the existing TAQA benchmark provided "limited discriminative power" for their 6B models, with low F1 scores across the board. While this justifies the creation of KairosQA, it raises questions about whether the benefits of sequential training only show up on the authors' own benchmark.
4.	How much of the "recency peak" is due to the uniqueness of recent data versus the repetition of historical data in the shuffled set?

---

> ### Author Rebuttal · Authors · 2026-03-31
>
> We thank the reviewer for recognizing the soundness of our experimental design and the meticulous construction of the KairosQA dataset. We are pleased that the reviewer appreciated our insights into how data shuffling impacts temporal knowledge. We have worked actively to address the concerns raised.
>
> ---
>
> &nbsp;
>
> ### Q1. Mitigating relative forgetting
>
>
> We agree that mitigating forgetting during sequential training is a crucial next step of our line of research. We performed early experiments with model merging (detailed in the table below), which demonstrate that incorporating earlier checkpoints degrades performance on recent data, with limited recovery of older knowledge. This indicates that simple weight interpolation is insufficient to mitigate the forgetting induced by sequential learning.
>
> _Table: Cloze Task results on KairosQA for Sequential and Soup Models_
>
> | Model name | Soup Coefficients (2018–2025) | 2018 | 2019 | 2020 | 2021 | 2022 | 2023 | 2024 | 2025 |
> | :--- | :--- | :---: | :---: | :---: | :---: | :---: | :---: | :---: | :---: |
> | **Seq 2025** | [0, 0, 0, 0, 0, 0, 0, 1] | 32.1 | 31.5 | 32.5 | 32.8 | 33.5 | 36.7 | 37.4 | 38.2 |
> | **Uniform** | [1, 1, 1, 1, 1, 1, 1, 1] | 28.3 | 27.2 | 25.9 | 24.5 | 23.5 | 22.7 | 23.8 | 23.0 |
> | **Linear** | [0.1, 0.2, 0.3, 0.4, 0.5, 0.6, 0.7, 0.8] | 29.0 | 29.3 | 29.3 | 29.9 | 30.6 | 31.1 | 30.1 | 27.9 |
> | **Exponential** | [0.5, 1, 2, 4, 8, 16, 32, 64] | 28.5 | 26.9 | 25.3 | 24.7 | 23.5 | 22.9 | 23.9 | 22.8 |
> | **One Every Two** | [0, 1, 0, 1, 0, 1, 0, 1] | 28.5 | 26.9 | 25.3 | 24.7 | 23.5 | 22.9 | 23.9 | 22.8 |
>
> &nbsp;
>
> We also tried a replay cool-down phase including data from previous years. This improved retention of past knowledge but simultaneously degraded performance on recent data (see our detailed response to hyVT, Q3). These preliminary experiments underscore that mitigating forgetting in sequential pretraining remains an open challenge and one we believe our release checkpoints and benchmark can help the community address.
>
> &nbsp;
>
> ### Q2. Snapshots Quality
>
>
>
> The fact that more recent Common Crawl (CC) snapshots yield better models was already observed in previous work, such as FineWeb [1]. To further validate this claim, we made the following additional experiment: starting from the 2021 sequential checkpoint, we performed the cooldown phase on data from the 2024 snapshot instead of the original 2021 data. This yielded a +1.5% improvement on OLMES (73.2% vs 71.7%), matching the shuffled model performance with the same number of training tokens (73.1%).  We believe that this provides direct evidence that the lag is mostly due to snapshot quality variability, rather than being an inherent cost of the sequential learning paradigm.
>
> &nbsp;
>
> ### Q3. Generalization of KairosQA
>
> KairosQA was designed independently of our sequential training protocol: the strongest evidence being that our shuffled baseline follows the same performance trends as open-source models across various sizes (Fig. 10). Thus, the benchmark does not favour our method by construction. This independence is further confirmed by the fact that sequential model's performance remains capped by its parameter count, staying below Gemma 3 12B for every year prior to 2024. Regarding TAQA, Figs. 11-13 show that its limited discriminative power affects all evaluated models in the 4B-12B range, not just ours.
>
> &nbsp;
>
> ### Q4. Uniqueness of recent data
> We have to say that we are not certain we have fully captured the reviewer's question.
>
> Is the question about the creation of our KairosQA benchmark? It targets evolving knowledge by repeating questions across years while their answers change. This creates an adversarial setting where historical data distracts from recent facts. As the table shows, questions overlap significantly (>80%) between all year pairs. Consequently, the "recency peak" demonstrates the sequential model’s ability to prioritize the current distribution, whereas the shuffled baseline fails to resolve these conflicting temporal facts simultaneously.
>
> Is the question about the fact that knowledge from older years (eg. 2018) is also present in a more recent snapshot (eg. 2024), while the inverse is not true? If so, this means that in the shuffled set, older knowledge is more represented that more recent knowledge, which we agree probably explains the patterns observed in temporal acquisition of knowledge in models trained on shuffled data.
>
> &nbsp;
>
> _Table: Proportion of Common Questions (Relative to Total in Row Year)_
> | | 2018 | 2019 | 2020 | 2021 | 2022 | 2023 | 2024 | 2025 |
> | :--- | :---: | :---: | :---: | :---: | :---: | :---: | :---: | :---: |
> | **2020** | 89 | 93 | 100 | 94 | 91 | 86 | 80 | 75 |
> | **2025** | 86 | 90 | 92 | 94 | 95 | 94 | 94 | 100 |
>
> —
> &nbsp;
>
> [1] [FineWeb]( https://huggingface.co/spaces/HuggingFaceFW/blogpost-fineweb-v1)

---

> > ### Author Rebuttal · Reviewer_8jn7 · 2026-04-03
> >
> > I will keep the acceptance score.

---

### Decision · Program_Chairs · 2026-04-30

**Decision:**

Accept (regular)

**Comment:**

This paper studies how the temporal ordering of pretraining data affects LLMs’ ability to acquire and retain time-sensitive knowledge, introducing a new benchmark (KairosQA) and showing that sequential training improves factual freshness without hurting general performance. Reviewers agree the paper has a strong and well-controlled experimental design that cleanly isolates data ordering effects, along with a valuable and carefully constructed benchmark that addresses an important and underexplored problem. The findings are novel, broadly relevant, and supported by extensive evaluation and convincing rebuttal experiments, making the work a solid contribution with clear community value.